# Enhanced automated detection of outbreaks of a rare antimicrobial-resistant bacterial species

**Yumiko Hosaka**[1]*, **Aki Hirabayashi**[1], **Adam Clark**[2], **Meghan Baker**[2], **Motoyuki Sugai**[1], **John Stelling**[2], **Koji Yahara**[1]*

**1** Antimicrobial Resistance Research Center, National Institute of Infectious Diseases, Higashimurayamashi, Tokyo, Japan, **2** Division of Infectious Diseases, Brigham and Women's Hospital, Boston, MA, United States of America

* hosaka-y@niid.go.jp (YH); k-yahara@niid.go.jp (KY)

**Data Availability Statement:** Anonymized data on the isolation and antimicrobial susceptibilities of VRE from the 15 hospitals are publicly available at http://dx.doi.org/10.6084/m9.figshare.27000574.

## Abstract

Surveillance of antimicrobial resistance (AMR) is a crucial strategy to combat AMR. Using routine surveillance data, we could detect and control hospital outbreaks of AMR bacteria as early as possible. Previously, we developed a framework for automatic detection of clusters of AMR bacteria using SaTScan, a free cluster detection tool integrated into WHONET. WHONET is a free software used globally for microbiological surveillance data management. We applied this framework to data from the Japan Nosocomial Infections Surveillance (JANIS), one of the world's most comprehensive and largest national AMR surveillance systems. Although WHONET-SaTScan has several cluster detection algorithms, no published studies have compared how different algorithms can produce varying results in cluster detection. Here, we conducted a comparison to detect clusters of vancomycin-resistant enterococci (VRE), which has been rare in Japan, by analyzing combinations of resistance to several key antimicrobials ("resistance profiles") using the comprehensive national routine AMR surveillance data of JANIS and validated the detection capabilities of each algorithm using publicly available reports of VRE clusters. All publicly reported VRE hospital outbreaks were detected as statistical clusters using the space-time uniform algorithm implemented in WHONET-SaTScan. In contrast, only 18.8% of the publicly reported outbreaks were detected using another algorithm (space-time permutation). The space-time uniform algorithm was also effective in identifying hospital wards affected by outbreaks attributed to specific resistance profiles. Although half of the publicly reported outbreaks were attributed to VRE resistant to five particular antimicrobials, four other resistance profiles also contributed to the outbreaks, highlighting the diversity of AMR bacteria within these occurrences. Our comparison revealed a clear advantage in using an algorithm (space-time uniform) for detecting VRE clusters in WHONET-SaTScan based on national surveillance data and further demonstrated the capability to distinguish detected clusters based on resistance profiles.

**Funding:** This work was supported by the Research Program on the challenges of Global Health issues from the Japan Agency for Medical Research and Development (AMED) (grant number 23jk0210040j0002). The funders had no role in study design, data collection and analysis, decision to publish, or preparation of the manuscript.

**Competing interests:** The authors have declared that no competing interests exist.

## Introduction

Antimicrobial resistance (AMR) is a critical global public health issue. As stated in the World Health Organization (WHO) global action plan on AMR in 2015, AMR surveillance is a crucial strategy for assessing the burden of AMR and improving the detection and control of emerging bacterial resistance and outbreaks locally, nationally, and globally [1]. In hospitals, infection control teams (ICTs) and medical microbiologists typically conduct AMR surveillance and evaluate and improve infection prevention and control (IPC) practices. However, many hospitals face difficulties in regularly and comprehensively monitoring the trends of potential AMR bacteria clusters due to a shortage of infection control practitioners (ICPs). Given this situation, the development of automated AMR outbreak detection based on routine surveillance data can be helpful in early identification of potential clusters of AMR bacteria.

Previously, we developed a framework for the automatic detection of clusters of AMR bacteria in hospitals [2] based on national AMR surveillance data in Japan. These data are stored in one of the largest AMR surveillance systems in the world, the Japan Nosocomial Infections Surveillance (JANIS), and covering the largest number of hospitals. As of January 2023, approximately 3000 hospitals were voluntarily participating in the JANIS Clinical Laboratory module, which collects all routine microbiological test results of both symptomatic and asymptomatic patients monthly through an online platform [3, 4]. Our previous study using JANIS database from 2015 to 2016 applied the cluster detection tool SaTScan [5], integrated into WHONET, a free microbiological surveillance data management software used in over 130 countries since 1989 [6], for the first time in an Asian country and across approximately 1000 hospitals. The framework using WHONET-SaTScan and the national AMR surveillance data in JANIS automatically detected clusters of AMR bacteria, many of which were validated and closely related to the findings from real-time practice [2].

In the aforementioned study, the statistical significance of a cluster was evaluated by Monte Carlo hypothesis testing, in which the space-time permutation algorithm was used to estimate the expected number of cases under the null hypothesis of the cluster. However, other algorithms are also available in the WHONET-SaTScan, and no published studies have compared how different algorithms can produce varying results in cluster detection. In this study, we compared the performances of two different algorithms for cluster detection of vancomycin-resistant enterococci (VRE) using the comprehensive national routine AMR surveillance data from JANIS. VRE are rare in Japan [7], but their incidence is sufficiently high to be monitored [8] using outbreak detection algorithms. We examined combinations of resistance to several key antimicrobials ("resistance profiles") [9] for detecting and distinguishing VRE clusters in WHONET-SaTScan. Additionally, we used publicly available reports on VRE clusters to validate the detection capabilities of each algorithm.

## Materials and methods

### Data preparation

All inpatient and outpatient data fields (e.g., facility code, encrypted patient ID, specimen ID, specimen reception date, specimen type, sex, date of birth, ward, department, and antimicrobial susceptibility testing results) for all specimens collected between January 2016 and December 2021 were extracted from the JANIS database, which stores both culture-positive and culture-negative diagnostic test and surveillance sample results, excluding data on antimicrobial usage. The data from the JANIS database were accessed for research purposes on 4 July 2023. Authors had no access to information that could identify individual participants during or after data collection. Data between 2016 and 2017 were used as historical baselines for

subsequent years, as described below. The percentage of hospitals voluntarily participating in JANIS was 23.1% of all 8,412 hospitals in 2018 and 26.7% of all 8,300 hospitals in 2021 across Japan [10]. The data of 1,896 hospitals were selected from the JANIS database because they had continuously provided JANIS with their microbiological data between 2018 and 2021. Hospitals with 200 or more beds showed a higher proportion of participation in JANIS than those with fewer than 200 beds (54.4% vs. 14.6%).

Given that VRE are generally rare in Japan, in the initial stage of data preparation, we extracted data from 142 hospitals in which the number of patients with VRE was ≥5 during the 4 years. For each patient with VRE, isolates not obtained during inpatient care were excluded from further analysis. The extracted data were converted to the WHONET format using a "JANIS to WHONET data parser" and imported into WHONET, a free software program developed and supported by the WHO Collaboration Centre for Surveillance of Antimicrobial Resistance.

## Selection of a core set of antimicrobials and hospitals

Given that antimicrobial susceptibility test (AST) practices between facilities were not identical, and not all antibiotics were tested on every isolate, we sought to ascertain a subset of antimicrobials that were relatively consistently tested across hospitals and over the study period. This subset of antimicrobials can be used to construct "resistance profiles" (combination of AST results) that could be compared between facilities and over time. We used WHONET to tabulate the number of susceptibility tests for each antimicrobial drug stratified by *Enterococcus* species in each of the 142 hospitals. Among several *Enterococcus* species, we focused on *E. faecium* in the subsequent analyses because this species constitutes the majority of VRE isolates in Japan and shows a higher rate of resistance to vancomycin (VAN) compared to *E. faecalis*. Specifically, rate of resistance to VAN in *E. faecium* is 2.6% (1,686 out of 65,363 isolates) for *E. faecium*, which is higher than 0.0063% (9 out of 142,300) for *E. faecalis* across 2,289 hospitals participating in JANIS in 2022 [10]. The high competency of resistance acquisition increases its impact as a causative pathogen of serious hospital-associated infections [11–14]. The JANIS surveillance system is based solely on phenotypical antimicrobial susceptibility testing. The automated antimicrobial susceptibility testing methods are mostly used for phenotypically testing vancomycin susceptibility in Japanese hospital laboratories. According to recent genomic surveillance in Japan (https://jarbs.net), only 0.93% of the phenotypically vancomycin resistant E. faecium did not agree with genotypic testing results. Hereafter, VRE will refer only to vancomycin-resistant *E. faecium* in this study. Postprocessing the tabulated results using WHONET identified a core set of five antimicrobials, all of which were tested for *E. faecium* in 96 out of the 142 hospitals: VAN, teicoplanin (TEC), levofloxacin (LVX), minocycline (MIN), and erythromycin (ERY). Besides VAN, the selection of the other four antimicrobial drugs was based on specific considerations. Teicoplanin, another glycopeptide, is commonly used as an alternative to VAN for treating VRE cases with renal dysfunction in Japan. Regarding LVX, enterococci acquire high-level resistance to quinolones through several mechanisms [12]. Regarding ERY, resistance to macrolides is widespread among enterococci [13]. MIN resistance can be transferred via specific transposons and varies among strains [15]. Moreover, all five antimicrobial drugs were tested for *E. faecium* in the majority of hospitals, and we expected the inclusion of TEC, LVX, MIN, and ERY to be useful in discriminating VRE isolates in terms of AST results and resistance profiles.

The JANIS surveillance system receives all raw testing results from the participating laboratories, and the CLSI guidelines were used to interpret the results of antimicrobial susceptibility testing and to categorize each isolate as susceptible or resistant to these antimicrobials [16].

Among the 96 hospitals that routinely tested the core set of five antimicrobials, we focused on 38 hospitals with ≥10 patients with VRE during the study period in the following analyses because hospitals with less than 10 patients with VRE did not have any publicly available official reports of VRE outbreaks required for validation of detected VRE clusters.

## Detection of statistical clusters with VRE using WHONET-SaTScan

Using WHONET, deduplication was conducted to create a data subset for subsequent analysis by selecting only the first VRE isolate showing each resistance profile of a given bacterial species per patient per surveillance period. The total number of patients with VRE was tabulated monthly during the study period at each hospital. The number of VRE isolates showing each resistance profile to the five core antimicrobials was also tabulated monthly for each hospital.

The batch version of the SaTScan software implemented within WHONET was used to detect statistically significant clusters of patients from whom isolates of the same bacterial species with identical resistance profiles were isolated [17]. SaTScan uses scan statistics that offer a relatively simple approach for determining whether the number of cases reported for a certain period is excessive [18]. We used the simulated prospective mode of WHONET-SaTScan, an iterative series of prospective analyses conducted over the length of the study period. This approach is not designed to detect longitudinal trends in baseline rates but to detect statistically significant short-term clusters suggestive of outbreaks meriting immediate response. In this study, the statistical significance of a cluster was evaluated by Monte Carlo hypothesis testing using both the space-time permutation and space-time uniform algorithm. The former has been included in the first version of an integrated WHONET-SaTScan package and is regarded as a helpful algorithm for the early detection of disease outbreaks because the probability model does not require a population-at-risk, as case data with information about spatial location and time are available [19]. The latter was developed for WHONET data interpretation and designed for case data with relatively consistent baseline case numbers of time [20]. We did not use the discrete Poisson model because it requires "Population data", meaning that the patient population in each geographic unit is needed for cluster detection based on geographical data. Since many hospitals do not have static patient population data, the Poisson-model approach is not applicable for cluster detection in this study.

The parameters for running WHONET-SaTScan were selected based on previous studies [2, 21–24]. The maximum cluster length cut-off 180 days was set, corresponding to the maximum temporal scanning window size. The statistical likelihood that the observed clusters are due to chance alone is expressed as the recurrence interval, equal to the inverse of the p-value [21]. The baseline period parameter (the number of days of historical data before the analysis) was set to 730 days, *i.e.* 2 years. We confirmed in advance the absence of large VRE outbreaks during the baseline period among the 15 hospitals we focus on below. The number of Monte Carlo simulations for calculating the recurrence intervals was set to 9,999. A recurrence interval threshold (a cutoff parameter in WHONET to filter out clusters detected by SaTScan) of 30 days was initially used in this study to detect clusters that one could expect to see by chance alone at most 12 times per year for further examination and validation. A recurrence interval threshold of 365 was subsequently used to check the robustness of the sensitivity and positive predictive values.

Statistical cluster detection was initially conducted at the hospital level, followed by the ward level. As ward data were not mandatory inputs for the JANIS database, cluster detection at the ward level was only possible in hospitals that provided ward data. For cluster detection at the ward level, patients with VRE were divided according to ward and resistance profiles. The analyses were implemented using WHONET macros and conducted separately for each hospital.

## Validation of detected VRE clusters using public reports

After statistical cluster detection using the two different scan statistical algorithms (space-time uniform and space-time permutation), we compared the detected clusters of VRE with publicly available official reports of VRE outbreaks between 2018 and 2021 in each hospital. The Infectious Disease Control Law in Japan obliges physicians to report every symptomatic case of VRE to a public health center which then sends these reports to the Ministry of Health, Labour and Welfare. Only aggregated data on the number of patients are regularly made public. However, there is no guarantee against underreporting. The absence of public reports of outbreaks in many hospitals with relatively high numbers of VRE patients does not necessarily indicate that no hospital outbreaks occurred in those hospitals. Generally, for the hospitals without publicly available VRE outbreak reports, we cannot make sure whether: 1) they had no outbreaks; 2) they had known outbreaks, but no reports are available; or 3) they had unrecognized outbreaks.

Public information sources included websites of each hospital, research papers, and newspaper articles. Hospitals with unconfirmed VRE outbreak timelines from any information source were excluded from the comparison of statistically detected clusters in order to calculate sensitivity and positive predictive value of cluster detection using WHONET-SaTScan. Without publicly available reports, it was impossible to classify detected clusters into true and false-positive detections. We could not use the publication date of outbreak report as a proxy for the start date of the outbreak because the outbreak report is often published several weeks after the exact outbreak start date.

## Ethics

Data were de-identified at each hospital and submitted to JANIS. The requirement for informed consent was waived by the Japanese Ministry of Health, Labor and Welfare (approval number 1108–5), and anonymous retrospective data stored in the JANIS database were exported and analyzed based on the approval according to Article 32 of the Japan's Statistics Act.

## Results

### Statistical clusters detected by resistance profiles

Through the process of selecting target hospitals (Fig 1), 15 hospitals that have been participating in JANIS since 2016 or earlier were chosen for the automated detection of statistically significant clusters after excluding 23 hospitals lacking publicly available information on either outbreak source or outbreak start date ("Information source of official VRE outbreak" in S1 Table).

All 15 hospitals exhibited two or more resistance profiles between 2018 and 2021 (S2 Table), enabling statistical cluster detection for each resistance profile compared to others observed over time with SaTScan. Altogether, six resistance profiles were identified across 15 hospitals (S2 Table), with five identified as forming statistically significant clusters in at least one hospital (Table 1). For example, resistance profile 1 (RP-1) in Table 1 and S2 Table displayed resistance to four antimicrobial drugs, excluding MIN, and accounted for 70.4–88.1% of patients with VRE in each of the top four hospitals by VRE patient count (A-1, C-1, D-1, B-2).

Table 1 shows the number of hospitals where clusters were detected for each of the five numbered resistance profiles using the space-time permutation and space-time uniform algorithms. For each resistance profile, the number of hospitals where clusters were detected using the space-time uniform algorithm was always greater than that using the space-time

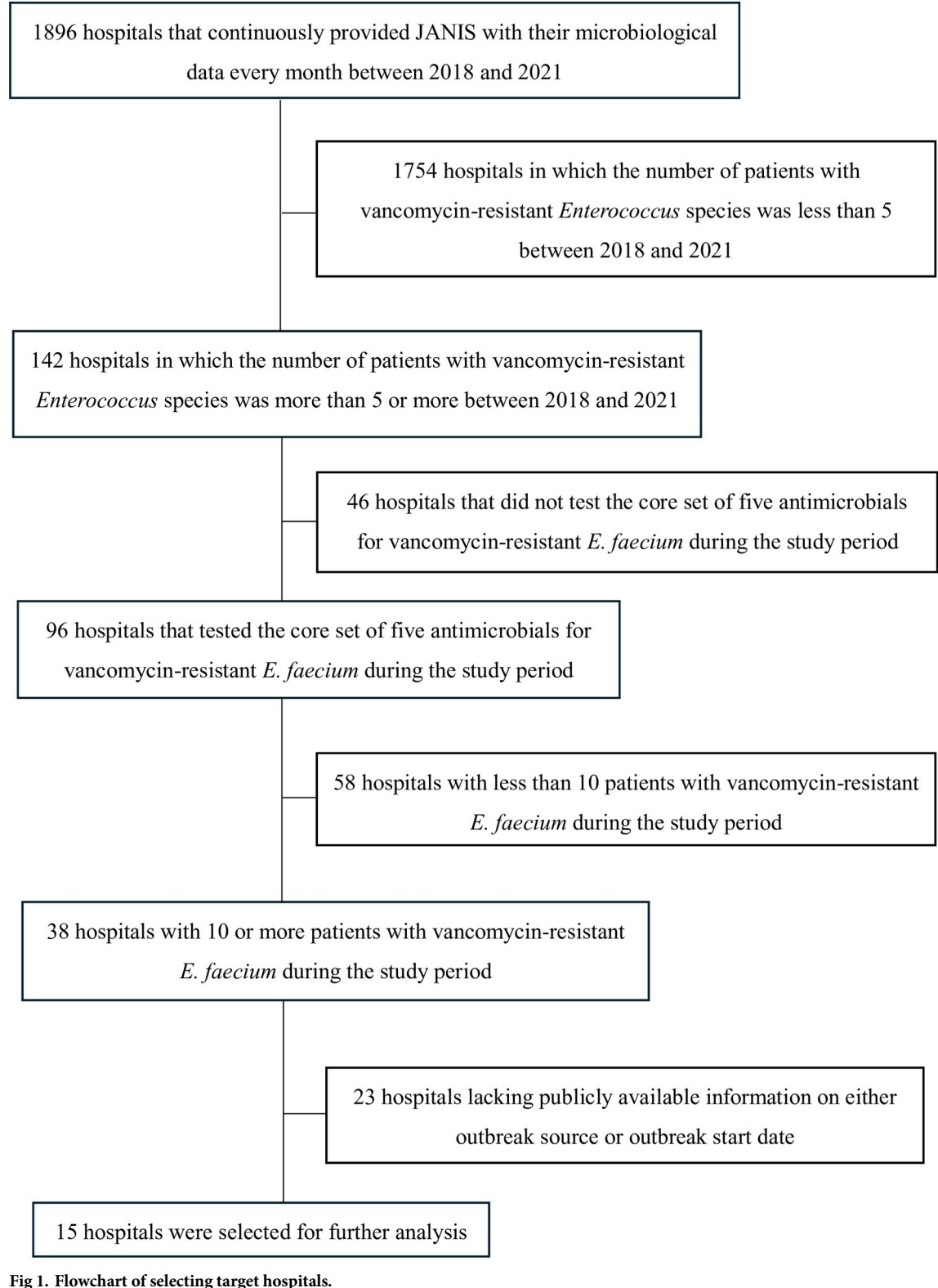

**Fig 1. Flowchart of selecting target hospitals.**

**Table 1. Five resistance profiles identified as forming statistically significant clusters in at least one hospital using the space-time permutation or space-time uniform algorithms.**

| RP number | Antimicrobial susceptibility test result | | | | | Number of hospitals with statistical clusters* | | Total number and % of isolates with each RP | Range of number of patients for the clusters detected by STU (minimum–maximum) |
|---|---|---|---|---|---|---|---|---|---|
| | VAN | TEC | LVX | MIN | ERY | detected by STP | detected by STU | | |
| RP-1 | R | R | R | S | R | 2 (2) | 11 (19) | 350 (50.9%) | 2–57 |
| RP-2 | R | R | R | R | R | 2 (2) | 9 (11) | 158 (23.0%) | 2–36 |
| RP-3 | R | S | R | S | R | 6 (6) | 8 (10) | 119 (17.3%) | 2–23 |
| RP-4 | R | S | R | R | R | 3 (3) | 7 (8) | 45 (6.6%) | 2–7 |
| RP-5 | R | R | R | R | S | 0 (0) | 1 (2) | 9 (1.3%) | 3–5 |
| RP-6 | R | R | R | S | S | 0 (0) | 0 (0) | 6 (0.9%) | 0 |

* The number of detected clusters is shown in parentheses

RP; resistance profile

STP; space-time permutation

STU; space-time uniform

The results of the antimicrobial susceptibility testing are expressed as R (resistance) or S (susceptibility) for each antimicrobial (VAN for vancomycin, TEC for teicoplanin, LVX for levofloxacin, MIN for minocycline, and ERY for erythromycin) within each RP.

RP numbers are listed in descending order of total number and % of isolates with each RP.

permutation algorithm, and all clusters detected by the space-time permutation algorithm were also detected by the space-time uniform algorithm. The number of clusters of the RP-1 mentioned above (R(VAN), R(TEC), R(LVX), S(MIN), R(ERY)) detected by the space-time uniform algorithm was 350 (50.9% of the total number of clusters) among 11 out of the 15 hospitals (73.3%).

## Comparison between statistical clusters detected using WHONET-SaTScan and actual outbreaks reported in public records

Fig 2 shows the time course of the statistical clusters of the five resistance profiles detected using the space-time uniform and space-time permutation algorithms in each of the 15 hospitals and the onset of publicly reported outbreaks between January 2018 and December 2021. The space-time uniform algorithm detected statistical clusters (dark blue arrows in Fig 2) in all 15 hospitals, whereas the space-time permutation algorithm detected clusters in nine of the 15 hospitals (silver arrows in Fig 2).

Comparing the detected statistical clusters with 16 actual outbreaks reported in public reports, the space-time uniform and space-time permutation algorithms exhibited sensitivities of 100% (16/16) and 18.8% (3/16), respectively when we used an alert threshold of 30 days for the recurrence interval. When we changed the recurrence interval threshold (a cutoff parameter in WHONET, described in Materials and Methods) from 30 to 365 days, sensitivities of space-time uniform and space-time permutation algorithms were 93.8% (15/16) and 6.3% (1/16), respectively (S1 Fig). The onset of the actual outbreaks (highlighted by red arrows in Fig 2) consistently corresponded with the initiation of the statistical clusters detected using the space-time uniform algorithm (indicated by the left ends of the dark blue arrows in Fig 2). Meanwhile, among 50 and 14 clusters detected using the space-time uniform and space-time permutation algorithms, respectively, the positive predictive value was at least 44.0% (22/50) for the space-time uniform algorithm and 21.4% (3/14) for the space-time permutation algorithm. Of the 16 actual outbreaks, half were caused by resistance profile 2 (RP-2: resistant to all five antimicrobials), followed by the aforementioned RP-1, causing seven outbreaks. When we

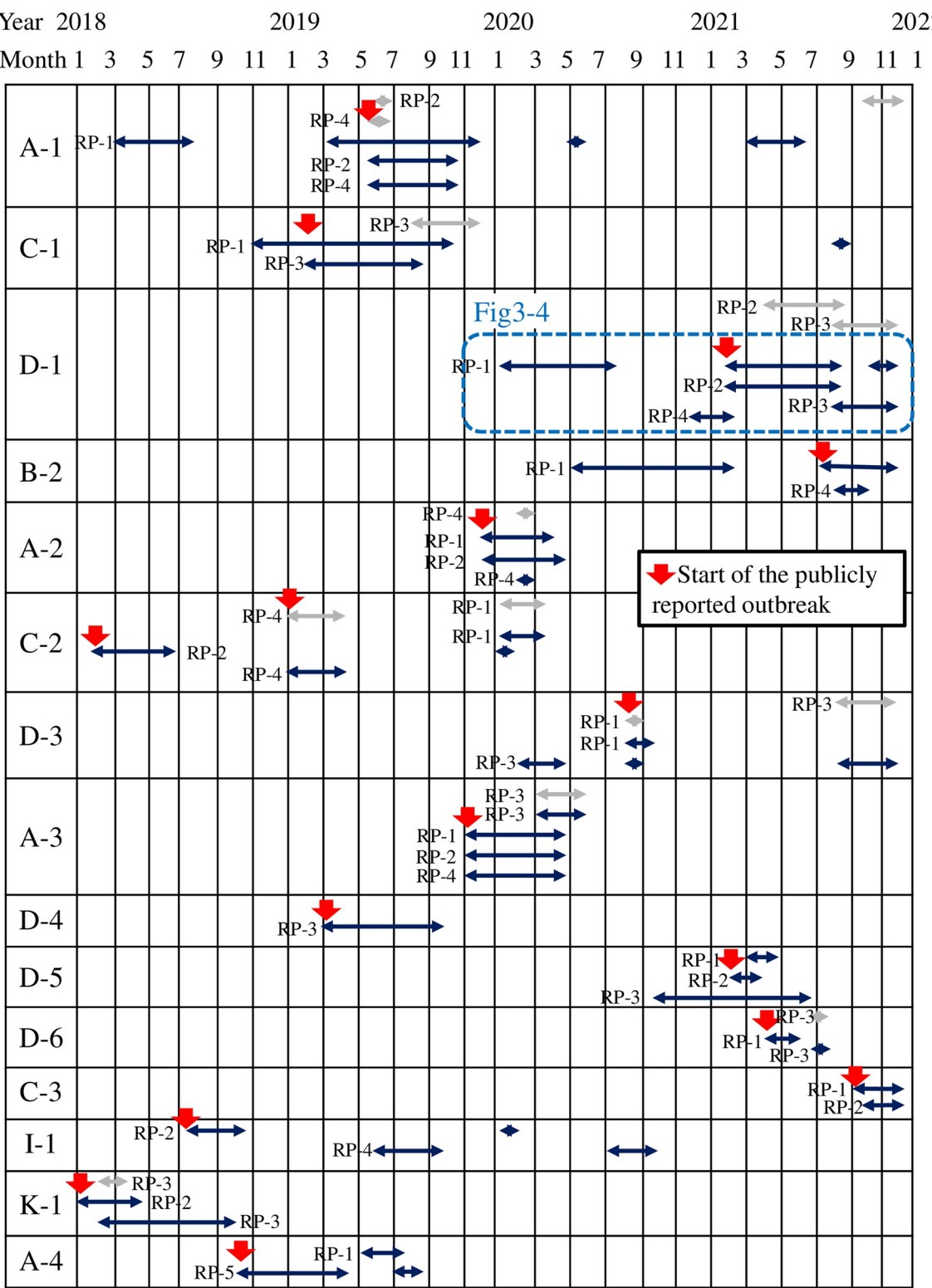

**Fig 2. Time course of statistical clusters detected using the space-time uniform and space-time permutation algorithms in each of the 15 hospitals and the onset of publicly reported outbreaks between January 2018 and December 2021.** The leftmost column represents each hospital code, which comprises a letter corresponding to the prefecture and a number denoting its order within that prefecture, as shown in S1 Table. Each hospital row shows the initiation and conclusion dates of statistical clusters for every resistance profile detected by WHONET-SaTScan and the onset of publicly reported outbreaks. The duration of the statistical clusters is

represented by arrows in light gray (detected using the space-time permutation algorithm) and dark blue (detected using the space-time uniform algorithm). Resistance profile numbers correspond to those listed in Table 1. Red arrows indicate the start of the outbreaks as publicly reported by various information sources. The blue dashed rectangle represents an outbreak detailed in Figs 3 and 4.

changed the recurrence interval threshold from 30 to 365 days, among 41 and six clusters detected using the space-time uniform and space-time permutation algorithms, respectively, the positive predictive value was at least 46.3% (19/41) for the space-time uniform algorithm and 16.7% (1/6) for the space-time permutation algorithm (S1 Fig).

### Further ward-level analysis of statistical clusters detected by space-time uniform algorithm in WHONET-SaTScan at a single hospital

Among the top three hospitals for the number of patients with VRE, further analysis was conducted for Hospital D-1 because Hospital C-1 lacked ward data, and Hospital A-1 had already published a detailed outbreak report as a research paper [8]. In Fig 3A, the monthly count of patients with VRE is displayed, while Fig 3B illustrates the number of patients for each of the four distinct resistance profiles alongside the duration of clusters (represented by dark blue horizontal lines and red vertical bars) detected using the space-time uniform algorithm.

The sudden increase in the number of patients with VRE coincided with the beginning of a publicly reported outbreak and the detection of two clusters associated with RP-1 and 2. This result indicates that the outbreak was not entirely clonal and comprised two types of isolates differing in susceptibility to MIN. Previously, the presence of a few patients with VRE with RP-1 each month between January and July 2020 was detected as a cluster using the space-time uniform algorithm. Additionally, a small number of patients with VRE with RP-3 and 4 during 3–4 months were detected as clusters.

Further ward-level analysis of Hospital D-1 is shown in Fig 4.

Ward A experienced only the statistical cluster of RP-1, whereas Ward B exhibited statistical clusters involving three types of RP-1, 2, and 3. In Wards A and B, the detected clusters of RP-1 coincided with the beginning of the publicly reported outbreak in February 2021, despite the difference in the length of the clusters. Another cluster of RP-1 detected in Ward A was responsible for the cluster at the hospital level (at the right in Fig 3) between October and November 2021. Before the beginning of the publicly reported outbreak, clusters of RP-1 were detected in Wards A (January–February 2020) and B (January–November 2020), potentially as early-stage signals of the actual outbreak.

### Discussion

A notable finding of this study was that all publicly reported VRE outbreaks were detected as statistical clusters using the space-time uniform algorithm implemented in WHONET-SaTScan. In contrast, only 18.8% of the publicly reported outbreaks were detected using the space-time permutation model. We confirmed the robustness of the results by changing the cutoff parameter of the recurrence interval threshold.

We previously conducted several studies for identifying clusters of infectious diseases using WHONET-SaTScan [17, 24], in which the space-time permutation model was used most frequently for detecting clusters of hospital-acquired microorganisms [21–23, 25]. However, only a few studies have compared multiple algorithms implemented in WHONET-SaTScan [20, 22]. To date, no study has compared the sensitivity of space-time permutation and space-time uniform algorithms for outbreak detection of AMR pathogens using national surveillance data. Given that the space-time uniform model was created specifically for WHONET [20],

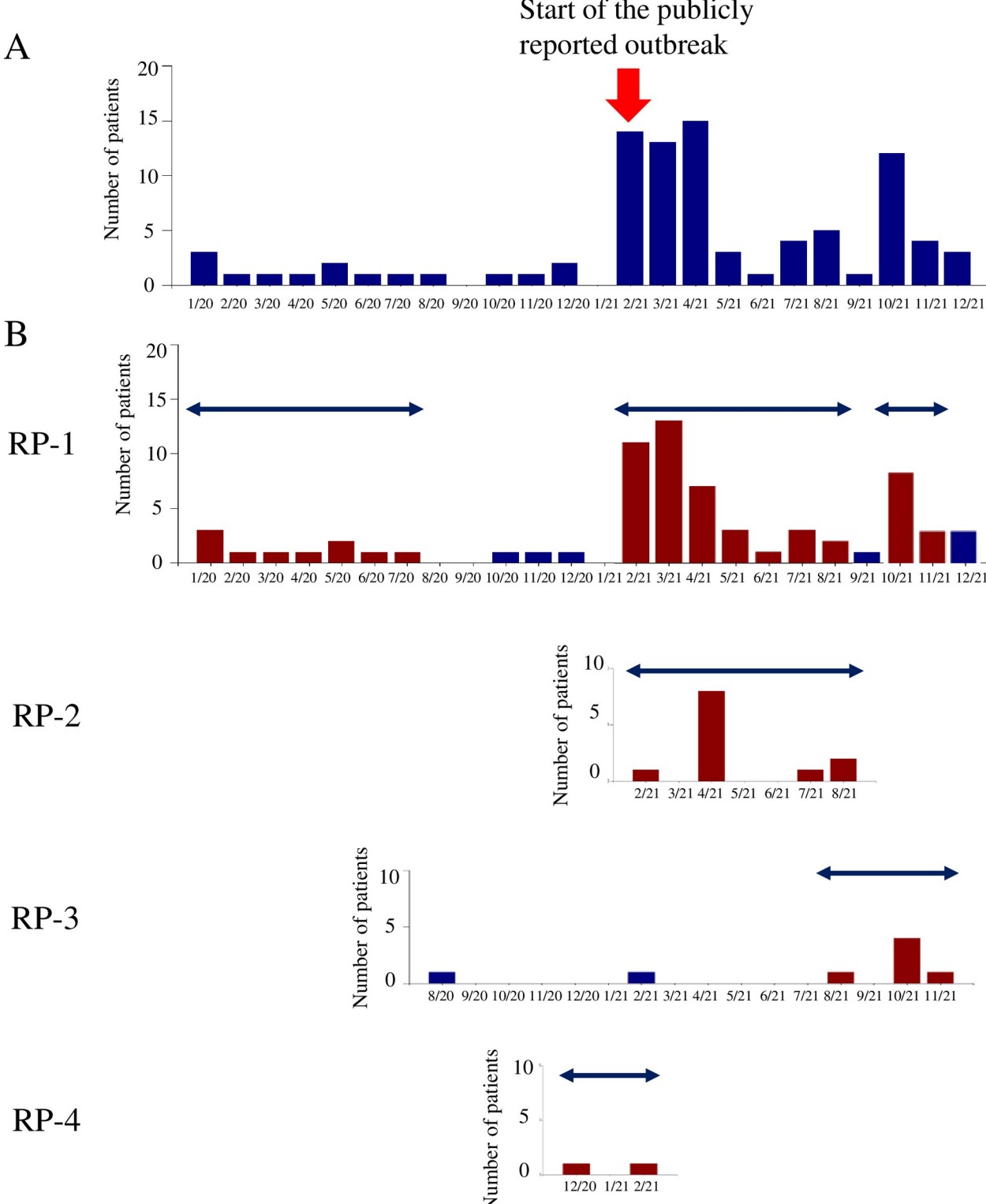

**Fig 3. Monthly trend between January 2020 and December 2021 in Hospital D-1.** (A) The number of patients with VRE. (B) The number of patients with each of four different resistance profiles. The dark blue horizontal arrows represent the duration of the statistical clusters detected using the space-time uniform algorithm, with monthly numbers shown as dark red vertical bars instead of blue. Resistance profile numbers correspond to those listed in Table 1. The red arrow indicates the start of the publicly reported outbreak.

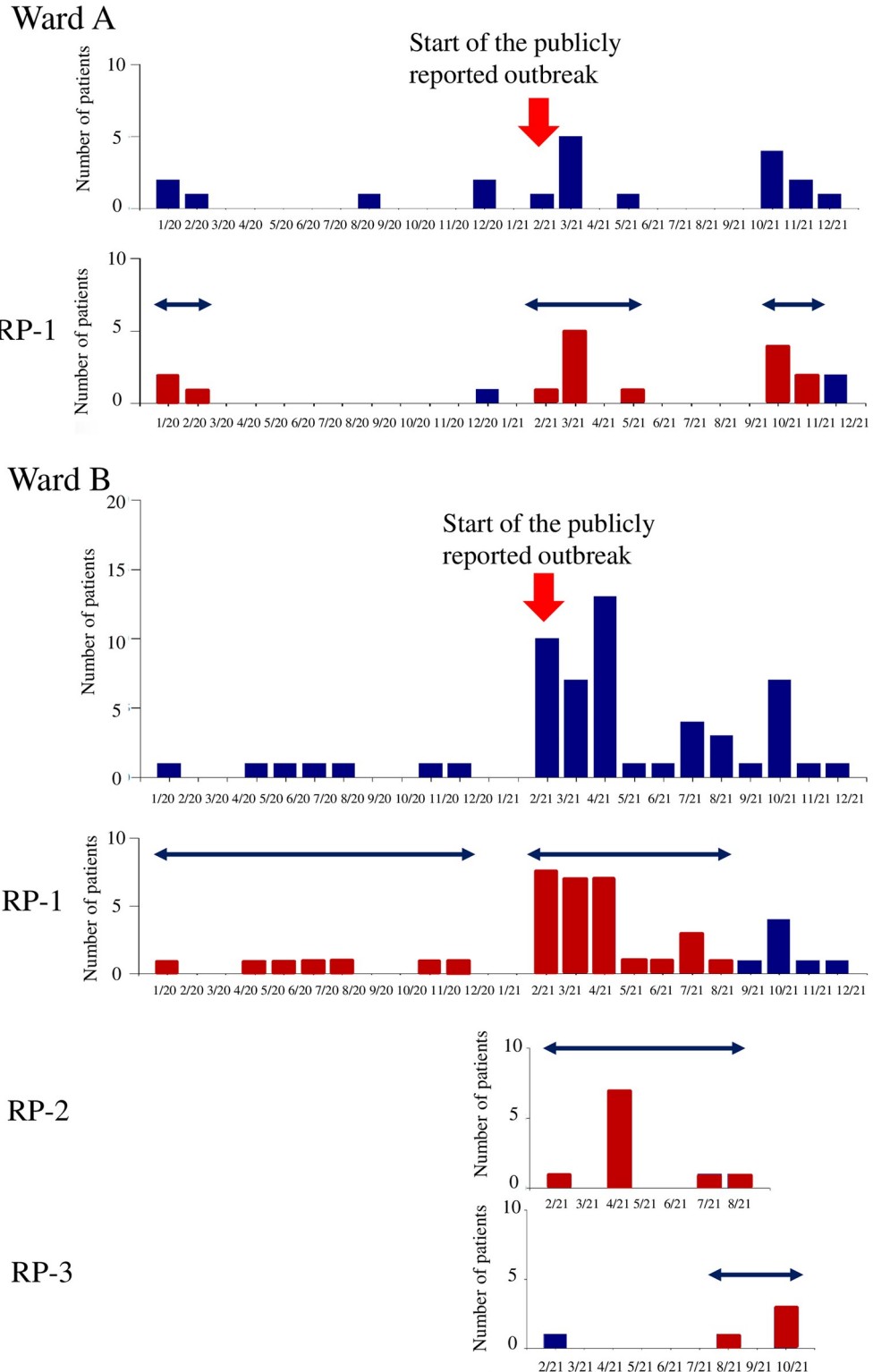

**Fig 4.** Monthly trend of the number of patients with VRE (upper) and that of the number of patients with each of the three resistance profiles in two different wards (Ward A and B) of Hospital D-1 between January 2020 and December 2021. The meanings of the arrows, bars, and RP numbers are the same as those in Fig 3.

more studies comparing the capability of AMR outbreak detection between the space-time uniform and other algorithms, such as space-time permutation, are warranted for improving automated AMR outbreak detection using WHONET-SaTScan. To the best of our knowledge, this is the first study to present a comparison focusing on VRE, a rare AMR pathogen, at least in Japan. However, we should keep in mind that the rarity of VRE cases also impacts infection control practices, in the sense that strong preventive measures are usually implemented as soon as a symptomatic VRE-positive case is observed. Early outbreak detection may be more important for other bacterial species than for VRE, because strong preventive measures might require the suspicion of an epidemic cluster for the former.

Regarding positive predictive value rather than sensitivity, a previous prospective study conducted at a United States (US) medical center indicated that an automated surveillance system based on the space-time permutation algorithm implemented in WHONET-SaTScan had a low positive predictive value for detecting high-risk transmission of pathogens (including several species) over 9 months; only 6 out of 45 high-risk clusters were detected as transmission events [26]. In the present study focusing on VRE, using the space-time uniform algorithm yielded a higher positive predictive value compared to that with the space-time permutation algorithm in the previous study. However, obtaining a precise sensitivity and positive predictive value is challenging because of the potential underreporting of actual outbreaks.

Regarding the numerous VRE clusters detected using the space-time uniform algorithm that were not associated with publicly reported outbreaks, several contrasting possibilities must be considered. First, the space-time uniform algorithm could also detect clusters that were not publicly reported, presenting a crucial advantage for early VRE cluster detection, as shown in Figs 3 and 4 as an example of Hospital D-1. Additionally, in Hospital A-1, a cluster was detected between March and July 2018 and another in March 2019, both predating the publicly reported outbreak that began in May 2019. This suggests the possibility of an unreported true initial outbreak in 2018, followed by a subsequent true outbreak in 2019. If the ICPs had responded to the first outbreak, the second outbreak might have been avoided. Similarly, if multiple statistical clusters were detected within a single ward or in a group of similar or adjacent wards, it could indicate a true outbreak in that ward or group of wards, even if it was not publicly reported. However, the detected clusters could be false positives. Alternatively, we must consider the possibility that the local ICT might have been aware of these case clusters and chose not to declare or report it as an outbreak. To disentangle these possibilities, a future study comparing the statistical clusters detected by space-time uniform and space-time permutation algorithms with actual outbreaks in real settings, based on prospective interviews with ICPs in each hospital, is warranted.

The primary difference between Figs 3 and 4 lies in the ease with which ICPs can identify cluster alerts of VRE with distinct resistance profiles in ward-level data. However, SaTScan may struggle to detect statistically significant clusters in the ward-level analysis. For instance, RP-4, which was initially detected as a statistical cluster at the hospital level (Fig 3B), disappeared in the analysis at the ward level (Fig 4). This was because there were two patients in two different wards, and the cluster detection using SaTScan was conducted for each ward separately. Namely, the sensitivity of the ward-level analyses decreases when the number of isolates becomes small due to dividing all isolates in a hospital by ward and resistance profiles.

In this study, we used combinations of resistance to several key antibiotics ("resistance profiles") to detect clusters of VRE by WHONET-SaTScan. The value of reporting and analyzing such 'full susceptibility profiles' rather than each of the antibiotics of interest has recently been highlighted [27]. Our study supports this approach and provides a model demonstrating the utility and effectiveness of analyzing full resistance profiles in detecting outbreaks, including

those that ICPs may have missed. Moreover, this study also aids in elucidating the diversity of AMR bacteria within these outbreaks, which might not be exclusively clonal (comprising a single dominant resistance profile) but rather heterogeneous, involving multiple resistance profiles. This approach can be used to monitor evolving microbial populations using routine phenotypic data as a practical and inexpensive proxy for genotypic characterization [9]. However, we must consider the reproducibility of the AST results for each of the selected antimicrobials for the resistance profiles. For example, the reproducibility of the AST results for TEC would affect the differences between resistance profiles 1 and 3, and those between 2 and 4, which were only based on whether they were susceptible or resistant to TEC.

Furthermore, we cannot conclude that VRE isolates with different resistance profiles by WHONET-SaTScan are genetically different because we did not conduct any molecular phylogenetic analysis of VRE. Relatedly, without molecular typing information, we cannot make sure whether all isolates within a cluster of a specific resistance profile are from the same strain. To address these issues, a strategy combining automated cluster detection using WHONET-SaTScan and in-depth molecular typing technologies, including whole-genome sequencing, is required, as previously suggested [28].

Most of our target patient population consisted of individuals colonized by VRE rather than those with VRE infections. This was because the target patients in the JANIS database were those from whom VRE were isolated from various samples. From a microbial and epidemiological perspective, including colonization results is valuable because it tells you that the microbe is moving and more patients are at risk. In a true outbreak where transmission can reach many patients, some will be colonized, and some will show symptoms with an infection diagnosis. Both are important with regards to outbreak detection and response. One limitation of using surveillance isolates is that a hospital may change their sampling practices over time, which can affect cluster detection. For example, you might have some short-term statistical "clusters" simply because the facility changes their surveillance strategy to doing more sampling. In addition, changes in laboratory testing practices for surveillance samples such as shift from culture screening on VRE plates to molecular screening with gene probes can also affect cluster detection.

This study had some other limitations. First, this was not a prospective study. The clinical significance of detected clusters was solely assessed based on public reports, which might have been affected by underreporting. Second, the criteria for collecting patient samples for culture and AST were not identical among the facilities. This is unavoidable in the JANIS system, wherein hospitals voluntarily provide specimen results using routine clinical samples. This could also affect the prevalence and proportion of resistance profiles tabulated from the JANIS data. Third, a significantly lower proportion of hospitals with fewer than 200 beds participated in JANIS compared to those with 500 or more beds (14.6% vs. 81.8%). Thus, larger hospitals may have a greater impact on the findings related to JANIS data than smaller hospitals. Fourth, only 15 hospitals with 10 or more VRE patients during the study period were included in the detailed analysis (as shown in the flowchart in Fig 1). This might be a problem in the generalization of our findings, for example among hospitals with fewer VRE patients and fewer publicly available reports on VRE outbreaks. Fifth, we did not have data on the exact number of true outbreaks because of the possibility of underreporting, which could bias the calculation of sensitivity.

## Conclusion

Despite these limitations, to our knowledge, this study is the first to demonstrate the clear advantage of using the space-time uniform algorithm for detecting VRE clusters in

WHONET-SaTScan based on national surveillance data and showcased the capability to distinguish detected clusters based on resistance profiles. Automated detection of VRE outbreaks is feasible for ICPs using standard computers and will prove valuable for early intervention in VRE infection control practices.

## Supporting information

**S1 Table. Each hospital code with number of patients with VRE, number of beds and information source of reported VRE outbreak.**
(DOCX)

**S2 Table. Detailed information about six resistance profiles identified across 15 hospitals.**
(DOCX)

**S1 Fig. Time course of statistical clusters detected using the space-time uniform and space-time permutation algorithms in each of the 15 hospitals and the onset of publicly reported outbreaks between January 2018 and December 2021 when the recurrence interval threshold in WHONET was changed from 30 to 365 days.**
(DOCX)

## Acknowledgments

The authors are grateful to all participating hospitals for their collaboration and for contributing their data to JANIS.

## Author Contributions

**Conceptualization:** Yumiko Hosaka, Meghan Baker, Motoyuki Sugai, John Stelling, Koji Yahara.

**Data curation:** Yumiko Hosaka, Adam Clark, Koji Yahara.

**Formal analysis:** Yumiko Hosaka, Adam Clark, John Stelling, Koji Yahara.

**Funding acquisition:** Yumiko Hosaka, Aki Hirabayashi, Meghan Baker, Motoyuki Sugai, John Stelling, Koji Yahara.

**Investigation:** Yumiko Hosaka, Aki Hirabayashi, Adam Clark, Meghan Baker, John Stelling, Koji Yahara.

**Methodology:** Yumiko Hosaka, Aki Hirabayashi, Adam Clark, Meghan Baker, John Stelling, Koji Yahara.

**Project administration:** Yumiko Hosaka, Aki Hirabayashi, Adam Clark, Meghan Baker, John Stelling, Koji Yahara.

**Resources:** Yumiko Hosaka, Adam Clark, Motoyuki Sugai, John Stelling, Koji Yahara.

**Software:** Yumiko Hosaka, Adam Clark, John Stelling, Koji Yahara.

**Supervision:** Yumiko Hosaka, Adam Clark, Meghan Baker, Motoyuki Sugai, John Stelling, Koji Yahara.

**Validation:** Yumiko Hosaka, Adam Clark, John Stelling, Koji Yahara.

**Visualization:** Yumiko Hosaka, Adam Clark, John Stelling.

**Writing – original draft:** Yumiko Hosaka, John Stelling, Koji Yahara.

**Writing – review & editing:** Yumiko Hosaka, Aki Hirabayashi, Adam Clark, Meghan Baker, Motoyuki Sugai, John Stelling, Koji Yahara.

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
