## [Decision Letter · Decision Letter 0]

10 Jul 2024

PONE-D-24-20250Enhanced automated detection of outbreaks of a rare antimicrobial-resistant bacterial speciesPLOS ONE

Dear Dr. Yahara,

Thank you for submitting your manuscript to PLOS ONE. After careful consideration, we feel that it has merit but does not fully meet PLOS ONE’s publication criteria as it currently stands. Therefore, we invite you to submit a revised version of the manuscript that addresses the points raised during the review process.

Please find the reviewers' comments below. Addressing all major issues raised by the reviewers is required for acceptance. Please provide a flowchart with reasons for exclusion of hospitals to improve transparancy. 

Please submit your revised manuscript by Aug 24 2024 11:59PM.  If you will need more time than this to complete your revisions, please reply to this message or contact the journal office at plosone@plos.org. Please include the following items when submitting your revised manuscript:A rebuttal letter that responds to each point raised by the academic editor and reviewer(s). You should upload this letter as a separate file labeled 'Response to Reviewers'.A marked-up copy of your manuscript that highlights changes made to the original version. You should upload this as a separate file labeled 'Revised Manuscript with Track Changes'.An unmarked version of your revised paper without tracked changes. You should upload this as a separate file labeled 'Manuscript'.

We look forward to receiving your revised manuscript.

Kind regards,

Luisa Anna Denkel

Academic Editor

PLOS ONE

Journal Requirements:

2. Thank you for stating the following financial disclosure: "This work was supported by the Research Program on the challenges of Global Health issues from the Japan Agency for Medical Research and Development (AMED) (grant number 23jk0210040j0002)."  

Reviewers' comments:

Reviewer's Responses to Questions

**Comments to the Author**

1. Is the manuscript technically sound, and do the data support the conclusions?

Reviewer #1: Yes

Reviewer #2: Partly

2. Has the statistical analysis been performed appropriately and rigorously? 

Reviewer #1: Yes

Reviewer #2: Yes

3. Have the authors made all data underlying the findings in their manuscript fully available?

Reviewer #1: Yes

Reviewer #2: No

4. Is the manuscript presented in an intelligible fashion and written in standard English?

Reviewer #1: Yes

Reviewer #2: Yes

5. Review Comments to the Author

Reviewer #1: The manuscript describes an interesting and relevant topic regarding the use of routine antimicrobial resistance surveillance data for the automated detection of within-hospital outbreak detection.

I think the paper describes a clear research question and a clear outline of the analyses and the, and the conclusion is sound and a direct response to the proposed research question.

Still, I had a number of issues and questions while reviewing the manuscript, which I would like to mention.

My biggest concern is the fact that a relatively small number of hospitals could be included in the analyses, compared to the seemingly large surveillance system with >3,000 participating hospitals. What does that say about the generalizability of the results and the practical implications for the use of Satscan? And: why were those hospitals excluded if no publicly available report of a VRE outbreak was found? The lack of a regular outbreak-reporting system makes it difficult to see publicly available reports on websites as a ‘gold standard’, but perhaps the situation in Japan is suitable for this. If this is the case, then this context should be explained more clearly in the introduction / discussion.

Some further comments / questions:

Introduction

- Lines 48-50: ‘In hospitals, infection control teams…. Prevention and control (IPC) practices.’

In some countries, AMR surveillance and evaluation is not primarily performed by (only) infection control teams, but by medical microbiologists (usually together with ICTs). Please also mention medical microbiologists here.

- Lines 77-79: ‘… certain hospitals have …… per hospital in recent years [8]’

The reference [8] seems not to be in place here. The sentence states that certain hospitals (plural) have reported clusters with a total number of VRE cases around 100. The paper in the reference describes one outbreak in one hospital with a total number of 60 VRE cases. Please add more references, or change the statement according to the information in the report referred to.

Material and methods

- Line 123:

Can you please explain what methods are mostly used for phenotypically testing vancomycin-susceptibility in the laboratories? What about the possibility of false-positive results of vancomycin-resistance? Is the JANIS surveillance system only based on phenotypical antimicrobial susceptibility testing, or are genotypic testing results (e.g. vanA or vanB testing) also available? Please describe this in the Material and methods section as well.

- Lines 135-137: ‘To categorize each….. antimicrobial susceptibility testing’

Does the JANIS surveillance system receive all raw testing results from the participating laboratories, and could those thus be reinterpreted for these analyses according to the CLSI guidelines? Or were the laboratory interpretations (by each separate participating laboratory) used for categorizing each isolate as susceptible or resistant?

- Lines 137-139: ‘Among the 96 hospitals…. the following analyses’

Why were only those hospitals included with 10 or more patients with VRE, and why not also those hospitals with 5-10 VRE during the study period?

- Lines 144-145: ‘showing each resistance profile of a given bacterial species per patient per surveillance period’

Only E. faecium was included, right? So it is always E. faecium as a species, or could other species be involved as well?

- Lines 189-195

What is common practice in Japan? Is each hospital outbreak normally reported in a public report, e.g. on the website of the hospital? Is it compulsory for hospitals to report a hospital outbreak in any publicly available report? What do you expect about underreporting? Many hospitals with relatively high numbers of VRE patients have no public report of an outbreak, does that mean that no hospital outbreak took place in those hospitals? There is one hospital with an outbreak which is reported in a research paper, but not on a website according to S1 Table. Please elaborate a bit more on clinical / routine practice in Japan.

Why were those hospitals with no publicly available official reports of VRE outbreaks during the study period excluded? I would be curious if Satscan detected clusters in any of these hospitals, and if you could find out if they concern real outbreaks, or only ‘false-positive detections’.

Results

- Line 209 / S1 Table:

The text says that 15 hospitals were chosen for the automated detection, but in S1 Table, only 13 hospitals are highlighted in grey.

- Line 210: ‘after excluding 23 hospitals’

Why were these hospitals excluded? If outbreaks were publicly reported without an exact outbreak start date, you could use the date of publication as a proxy? And even if there was no publicly available information on an outbreak, why would that hospital be excluded? Perhaps no real outbreak ever took place in that hospital?

- Line 215-216: ‘Satscan, which requires a comparison of multiple resistance profiles in each hospital.’

I don’t really understand this. Why does Satscan require multiple resistance profiles in each hospital? What if an outbreak happens in a hospital with only 1 particular VRE strain with isolates all having the same resistance profile? Is in that case Satscan not able to detect a cluster? Please rephrase or explain what is meant in this sentence.

- Line 227-229: ‘Clusters of the RP-2…. In 73.3% of the 15 hospitals.’

Why is the percentage (73.3%) of the 15 hospitals depicted in which the clusters of the RP-2 were detected? I would be more interested in the percentage of the total number of clusters (in how many hospitals) instead of the percentage of the total number of hospitals.

- Table 1

Why is the total number of isolates with each resistance profile relevant? The number of isolates strongly depends on the majority of a specific outbreaks (in this the majority of all outbreaks with a specific RP), which depends on the infection prevention measures and possibility to contain the outbreak. The number of isolates for each outbreak is interesting (it shows how ‘sensitive’ the cluster detection is), but it is not really interesting to know the total number overall for each RP, I think. Perhaps you could add the range of number of patients for the detected clusters (minimum – maximum), which says more about the clusters / outbreaks.

- Lines 249-252: ‘Comparing the detected statistical clusters….. for the recurrence interval.’

You don’t have a gold standard on the exact number of outbreaks, because of the possibility of underreporting, and thus, strictly, you cannot use sensitivity calculations.

- Did you compare the numbers of patients with VRE within clusters detected by Satscan with the numbers of positive patients reported in the publicly available reports? Were those number (more or less) comparable?

- Lines 298-300: ‘Previously, the presence of a few….. space-time uniform algorithm.’

Do you know anything of genotyping results of these RP-2 VRE-isolates of those few patients? Could it be possible that, although they have a similar RP, that they comprise different VRE strains?

Was this information used for feedback to the hospital, and was additional information from the hospital requested, if the outbreak might have started earlier than the publicly available starting date?

Discussion

- Lines 356-357: ‘However, obtaining a precise….. of actual outbreaks.’

This also accounts for sensitivity.

- Lines 363-366: ‘This suggests the possibility of an unreported true initial outbreak in 2018, followed by a subsequent true outbreak in 2019.’

See my previous question with lines 298-300: Has there been any communication with Hospital A-1 about the findings of the cluster detection? Were the ICPs possibly already aware about the positive findings in 2018 and March 2019 (without having reported publicly)?

- Lines 377-383:

I agree, ward-level analyses does not always help in improved cluster detection, for example because of patients being easily transferred between different departments, or healthcare workers working on multiple departments. Could you add something about your conclusion on that? What is your recommendation?

- Lines 392-402:

And additionally: a similar resistance profile does not mean that multiple VRE isolates comprise one similar strain. Thus, how sure are you that all isolates within a cluster with e.g. RP-2 are from the same strain. It seems that RP-2 is a dominant resistance profile, and thus different strains will have a similar (this particular) resistance profile, and a finding of more isolates with RP-2 does not necessarily mean that there is transmission between individuals or an outbreak event.

- Limitations: in the introduction and first part of Material and methods, the large coverage of the JANIS surveillance system is described as a strength, with a participation of around 8,300 hospitals. In the end, only 15 hospitals are included in the analysis for this paper, which is obviously only a very small part of the total number of participating hospitals, but still also a relatively small part of the 142 hospitals with >5 VRE during the study period 2018-2021, or of the 96 hospitals where all 5 selected antimicrobial were tested for E. faecium. Please describe this in the limitations, and perhaps add why you think this might (or not) be a problem in the generalization of your findings.

- Could this algorithm in the Satscan software also be used for multi-institutional outbreak detection? How does that work with the space-time uniform algorithm? Or is this not possible, thus how can Satscan be of practical use? And could you add that to the discussion?

Reviewer #2: I would like to thank the authors for their work on this important aspect of infection control. Indeed, I agree that surveillance is a key part in AMR management and that automated algorithms may help IC team in this time-consuming and often tedious task.

As noted in the manuscript, VRE cases are a good candidate for the research question at hand, as they are sufficently rare for outbreaks to be noted and publicly reported. However, the fact that VRE cases are rare also has an impact of infection control practices, in the sense that strong preventive measures are usually implemented as soon a VRE positive sample is observed. Early outbreak detection is thus less critical than for other bacterial species for which strong preventive measures might require the suspicion of an epidemic cluster.

The main result of the manuscript is that the continuous uniform model of SaTScan performs better than the space-time permutation model, i.e. that it can detect more outbreaks. This is not an entirely new result (see Master’s thesis by M. Bokhari) but it has to my knowledge not been studied at this scale for hospital outbreaks. There is another algorithm, however, that sometimes shows an even higher sensitivity thhat the continuous uniform model, which is the discrete Poisson model. Can you explain why it was not mentionned or tested?

One of the main issue I had reading the manuscript is that the overall description of the selection process is hard to follow: we learn line 106 that isolates from outpatients were excluded, line 137, that hospitals that don’t test for the antimicrobials chosen for the resistance profiles were excluded, as were hospitals with less than 10 VRE patients during the study period. We then learn line 182 that only data from 2018 to 2021 were used, and we finally learn lines 209-212 that 23 other hospitals were excluded because they didn’t have publicly available information about VRE outbreaks. I suggest gathering of this information in the same paragraph or section and adding a flow diagram for more clarity.

Some technical aspects of the method should in my opinion be clarified:

- Line 168; the meaning of the maximum temporal window size could be explained in more details for readers that are not well versed in this very specific statistical framework.

- Lines 169-170; I may have misunderstood the sentence, but how can a likelihood be expressed in days?

- The reccurrence interval threshold was set to 365 days for the sensitivity analysis; does it mean that only one outbreak per year could be expected? It seems very different from the "one outbreak per month" scenario that was used in the main analysis. Can you explain this choice theoretically or epidemiologically?

- Baseline data for outbreak detection need to be in statistical control, meaning that there is no outbreak during this periode. Did you check for that and if so, how?

I have no particular comment regarding the antibiotics chosen for the resistance profiles, but the reasons given for their inclusion are difficult to understand at first read. I understand that TEC was chosen because it is an alternative to VAN, but why is it relevant in a context of surveillance to monitor this specific resistance? for LVX, MIN and ERY, I believe they were chosen because resistance to these antimicrobials are supposed to be more frequent, but again it is not clear as to why they are interesting for surveillance purposes. Are these resistances interesting for clinical or public health reasons? Maybe they are useful because of their discriminant value?

The part of the manuscript that explore the results by wards in hospital D-1 is out of the scope of this study and doesn’t add anything to the research question. It should probably be a manuscript of its own.

Finally, you will find below some minor issues that I believe should be addressed:

- line 62 "A previous study ..." please be more specific. Several studies have in fact used SaTScan, even within the WHONET application, to monitor hospital outbreaks. It is important for the reader to understand from the start that the study that your are talking about is related to your team and the JANIS database.

- line 70 "In a previous study ..." Please be more specific. Is it the same as the study mentionned line 62?

- line 75 "we conducted such a comparison to detect clusters ...". I believe the sentence is misleading. The comparison was not made to detect clusters, but cluster detection performances were compared between two algorithms.

- lines 98-102: A more succint description of the JANIS hospitals would suffice, details on the distributions are in my opinion superfluous.

- Lines 123-124: percentages would be more readable that fractions.

- Line 126: VR E. faecium are already VREs, I guess what you really meant is that VREs will refer to E. faecium only.

- line 174-175: A statistically significant signal does not equal to a real outbreak indeed, but I don’t see the logical relationship of this with the start of the sentence that relate to the minimal size of the cluster.

- Lines 198-201: Although it is easily deducible, I suggest precising the country in the Ethics paragraph ("... the Japanese Ministry of Health ...", "...Japan’s Statistics Act" for more clarity.

- lines 206-208: these pieces of information are already stated in the methods section.

- The labelling of the resistance profiles doesn’t seem to follow any obvious scheme. Labelling them, for example, in order of their frequency might be more logical for the reader.

- For small sample sizes, adding the count is sometimes as useful as the proportion, and I suggested adding them to the result. Line 229 for example would be "... 73.3% (n=11) of the 15 hospitals".

- The results should be in the past tense (e.g. line 251 "... when we useD...").

- Lines 260-261: the remark about the potential under-reporting is an element of discussion and not be in the results.

- Line 258: I didn’t understand what "among 50 and 13 clusters detected" related to. Same remark for lines 263-264.

6. PLOS authors have the option to publish the peer review history of their article (what does this mean?). If published, this will include your full peer review and any attached files.

Reviewer #1: No

Reviewer #2: No

---

## [Decision Letter · Decision Letter 1]

10 Sep 2024

PONE-D-24-20250R1Enhanced automated detection of outbreaks of a rare antimicrobial-resistant bacterial speciesPLOS ONE

Dear Dr. Yahara,

Thank you for submitting a revised version of your manuscript to PLOS ONE. After careful consideration, we feel that it has merit but does not fully meet PLOS ONE’s publication criteria as it currently stands. Therefore, we invite you to submit a revised version of the manuscript that addresses the points raised during the review process. One reviewer has some minor issues (point 6 comments to the authors, see below) that need to be addressed before publication. 

Please submit your revised manuscript by Oct 25 2024 11:59PM If you will need more time than this to complete your revisions, please reply to this message or contact the journal office at plosone@plos.org. Please include the following items when submitting your revised manuscript:

We look forward to receiving your revised manuscript.

Kind regards,

Luisa Anna Denkel

Academic Editor

PLOS ONE

Journal Requirements:

Reviewers' comments:

Reviewer's Responses to Questions

**Comments to the Author**

1. If the authors have adequately addressed your comments raised in a previous round of review and you feel that this manuscript is now acceptable for publication, you may indicate that here to bypass the “Comments to the Author” section, enter your conflict of interest statement in the “Confidential to Editor” section, and submit your "Accept" recommendation.

Reviewer #1: All comments have been addressed

Reviewer #2: All comments have been addressed

2. Is the manuscript technically sound, and do the data support the conclusions?

Reviewer #1: Yes

Reviewer #2: Yes

3. Has the statistical analysis been performed appropriately and rigorously? 

Reviewer #1: Yes

Reviewer #2: Yes

4. Have the authors made all data underlying the findings in their manuscript fully available?

Reviewer #1: Yes

Reviewer #2: No

5. Is the manuscript presented in an intelligible fashion and written in standard English?

Reviewer #1: Yes

Reviewer #2: Yes

6. Review Comments to the Author

Reviewer #1: (No Response)

Reviewer #2: Thank you for addressing our comments, I hope you found them useful to improve your manuscript.

I only have minor issues with this new version of the manuscript, which I detail below:

I don’t understand the text added lines 77-79. Because one hospital had high rates of VRE incidence, this makes it suitable for comparison? I think you mean that VREs are rare in Japan, but that their incidence is sufficiently high to be monitored using outbreak detection algorithms. I suggest rewriting this sentence for more clarity.

lines 123-126: this long part in parentheses containing sub parts in parentheses is are to read and contains a lot of redundancies. I again suggests rewriting this sentence for more clarity.

7. PLOS authors have the option to publish the peer review history of their article (what does this mean?). If published, this will include your full peer review and any attached files.

Reviewer #1: No

Reviewer #2: No

---

## [Editor Report · Decision Letter 2]

15 Sep 2024

PONE-D-24-20250R2Enhanced automated detection of outbreaks of a rare antimicrobial-resistant bacterial speciesPLOS ONE

Dear Dr. Yahara,

Thank you for submitting your revised manuscript to PLOS ONE. After careful consideration, we feel that it has merit but does not fully meet PLOS ONE’s publication criteria as it currently stands. Therefore, we invite you to submit a revised version of the manuscript that addresses the points raised during the review process.

Please follow the instructions suggested by one reviewer below. 

We look forward to receiving your revised manuscript.

Kind regards,

Luisa Anna Denkel

Academic Editor

PLOS ONE

Journal Requirements:

Reviewers' comments:

Thank you for addressing our comments, I hope you found them useful to improve your manuscript.

I only have minor issues with this new version of the manuscript, which I detail below:

I don’t understand the text added lines 77-79. Because one hospital had high rates of VRE incidence, this makes it suitable for comparison? I think you mean that VREs are rare in Japan, but that their incidence is sufficiently high to be monitored using outbreak detection algorithms. I suggest rewriting this sentence for more clarity.

lines 123-126: this long part in parentheses containing sub parts in parentheses is are to read and contains a lot of redundancies. I again suggests rewriting this sentence for more clarity.

---

## [Editor Report · Decision Letter 3]

8 Oct 2024

Enhanced automated detection of outbreaks of a rare antimicrobial-resistant bacterial species

PONE-D-24-20250R3

Dear Dr. Koji Yahara,

We’re pleased to inform you that your manuscript has been judged scientifically suitable for publication and will be formally accepted for publication once it meets all outstanding technical requirements.

Kind regards,

Luisa Anna Denkel

Academic Editor

PLOS ONE
---

## [Editor Report · Acceptance letter]

14 Oct 2024

PONE-D-24-20250R3 

PLOS ONE

Dear Dr. Yahara, 

I'm pleased to inform you that your manuscript has been deemed suitable for publication in PLOS ONE. Congratulations! Your manuscript is now being handed over to our production team.

Kind regards, 

on behalf of

Dr. Luisa Anna Denkel 

Academic Editor

PLOS ONE